# Convergence of immune escape strategies highlights plasticity of SARS-CoV-2 spike

Xiaodi Yu[1]☺, Jarek Juraszek[2]☺, Lucy Rutten[2], Mark J. G. Bakkers[2], Sven Blokland[2], Jelle M. Melchers[2], Niels J. F. van den Broek[2], Annemiek Y. W. Verwilligen[2], Pravien Abeywickrema[1], Johan Vingerhoets[3], Jean-Marc Neefs[4], Shah A. Mohamed Bakhash[5], Pavitra Roychoudhury[5], Alex Greninger[5], Sujata Sharma[1], Johannes P. M. Langedijk[2]*

1 Structural & Protein Sciences, Janssen Research and Development, Spring House, Pennsylvania, United States of America, 2 Janssen Vaccines & Prevention BV, Leiden, the Netherlands, 3 Janssen Pharmaceutica N.V., Clinical Microbiology and Immunology, Beerse, Belgium, 4 Janssen Pharmaceutica N. V., Discovery Sciences, Beerse, Belgium, 5 Department of Laboratory Medicine and Pathology, Virology Division, University of Washington, Seattle, Washington, United States of America

☺ These authors contributed equally to this work.
* hlangedi@its.jnj.com

**Data Availability Statement:** The maps were deposited in the electron microscopy data bank (EMDB) with IDs EMD-29455, EMD-29454, EMD-

## Abstract

The global spread of the SARS-CoV-2 virus has resulted in emergence of lineages which impact the effectiveness of immunotherapies and vaccines that are based on the early Wuhan isolate. All currently approved vaccines employ the spike protein S, as it is the target for neutralizing antibodies. Here we describe two SARS-CoV-2 isolates with unusually large deletions in the N-terminal domain (NTD) of the spike. Cryo-EM structural analysis shows that the deletions result in complete reshaping of the NTD supersite, an antigenically important region of the NTD. For both spike variants the remodeling of the NTD negatively affects binding of all tested NTD-specific antibodies in and outside of the NTD supersite. For one of the variants, we observed a P9L mediated shift of the signal peptide cleavage site resulting in the loss of a disulfide-bridge; a unique escape mechanism with high antigenic impact. Although the observed deletions and disulfide mutations are rare, similar modifications have become independently established in several other lineages, indicating a possibility to become more dominant in the future. The observed plasticity of the NTD foreshadows its broad potential for immune escape with the continued spread of SARS-CoV-2.

## Introduction

The surface spike (S) protein of SARS-CoV-2 is critical for the viral life cycle as it constitutes the main target of neutralizing antibodies [1–4] and is therefore at the basis of prophylactic vaccines. S is a large, trimeric glycoprotein that mediates both binding to host cell receptors and fusion of the viral and host cell membranes through its S1 and S2 subunits respectively [5–8] (S1 Fig). The S1 subunit comprises two distinct domains, an N-terminal domain (NTD) and a host cell receptor-binding domain (RBD), that are both targets of neutralizing antibodies with high neutralizing and protective potential [9]. The RBD transitions to the "up" position

29456, and the atomic models in the protein data bank (PDB) with IDs 8FU8, 8FU7, and 8FU9.

**Funding:** The author(s) received no specific funding for this work.

**Competing interests:** I have read the journal's policy and the authors of this manuscript have the following competing interests: J.J., L.R., M.J.G.B., and J.P.L. are co-inventors on related vaccine patents. X.Y., J.J., L.R., M.J.G.B., J.J.M., S.B., N.J. F.vdB., A.Y.W.V., P.A., J.V., J.N., S.S., and J.P.L. are employees of Janssen Vaccines & Prevention BV. X.Y., J.J., L.R., J.V., and J.P.L. hold stock of Johnson & Johnson.

for engagement with the cell-surface protein ACE2 which depends on the movement of NTD relative to S2 and the RBD [10,11]. The NTD possesses a galectin fold with five exposed protruding loops (N1-5) and can bind to sialic acid-containing moieties on host cell membranes although this binding specificity has been lost in recent lineages and the exact function of NTD remains unclear [12–14]. Despite the lack of understanding of NTD function and the neutralization mechanism of the NTD-specific antibodies, the domain is immunodominant and binds antibodies with high neutralizing and protective potential [2,9,15–22]. NTD contains a non-glycosylated immunogenic supersite centered around N1, N3, and N5 loops that is a hotspot for spike mutations, including deletions [9,23–26]. Besides the mutations and deletions, a novel escape mechanism was described for NTD by mutations in the signal peptide that cause a shift in the cleavage position up to Cys15 and result in the loss of a disulfide between Cys15 and Cys136 ($DS_{15-136}$) with unknown structural consequences [18,27,28].

In this study, we characterize spikes of two isolates, obtained from samples of infected individuals in Brazil (ΔN135) and Peru (ΔN25) in January 2021. Both S proteins contained multiple very large deletions within the NTD, much larger than in the current and past variants of concern (VoCs). Both ΔN135 and ΔN25 S proteins fold correctly and maintain fusion capacity despite the disulfide loss and large deletions in the NTD. The cryo-EM structure of ΔN135 for the first time shows the structural consequence of the loss of the DS15-136 disulfide and both spike structures reveal major remodeling of the NTD domains. The structural insights are corroborated by *in vitro* antigenicity profiling which underlines the potential impact of these spike modifications on immune escape.

## Results

Next-generation sequencing analysis of SARS-CoV-2 RNA isolated from nasal swab samples collected from study participants in the Phase 3 trial of the Ad26.COV2.S vaccine (VAC31518-COV3001, Ensemble, funded by Janssen Research and Development and others, ClinicalTrials.gov number NCT04505722 [29]) revealed various adaptations in the S gene sequences. Multiple study participants from Peru and one from Argentina infected with the C.37, lambda variant, showed common mutations in the NTD and the RBD and a unique large deletion of residues 63–75 in the N2 loop of the spike. Since the spike contains deletions in the N5 loop common for C.37 and N2, the spike has been named ΔN25 (Figs 1A and S1 and S1 Table). Samples obtained from two study participants infected with the B1.1.129 variant that were taken on January 12th and 17th of 2021 in Sao Paolo, Brazil, showed a number of identical amino acid S sequences, very different from the global consensus. Apart from several previously described RBD mutations, these sequences contained a modification in the signal peptide next to the N1 loop and two large deletions in the NTD, one in a β -strand preceding the N3 loop (residues 136–144), and another in the N5 loop (residues 258–264). Due to the location of these modifications in the NTD loops, the spike has been named ΔN135 (Figs 1A and S1 and S1 Table).

### Variant spikes remain fusogenic and allow cell entry

Given the extensive changes that ΔN25 and ΔN135 spikes had accumulated compared to the original SARS-CoV-2 strain, we endeavored to confirm their ability to successfully accomplish membrane fusion. We measured the impact of the changes in the full-length variant spikes on fusion activity compared with the wild-type Wuhan-Hu-1 (GenBank accession number: MN908947) in a cell-cell fusion assay that makes use of a fluorescent reporter protein to visualize syncytia formation [30]. HEK293 cells were transiently transfected with plasmids encoding S, ACE2, TMPRSS2, and GFP. Transfection of GFP alone, or of a prefusion-stabilized S

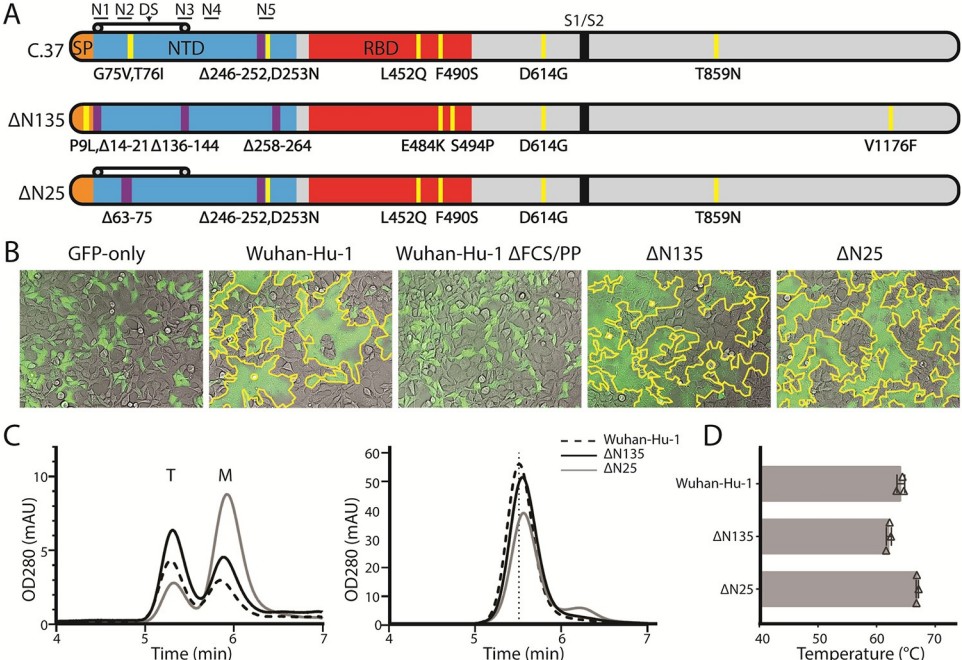

**Fig 1. Characterization of variant spikes. A.** Schematic representation of the C.37, ΔN135 and ΔN25 spikes with SP, NTD, RBD, and S1/S2 cleavage site in orange, blue, red, and black, respectively. Five supersites, mutations (yellow) and deletions (magenta) were indicated above and below the bar. **B.** Cell-cell fusion assay in HEK293 cells by co-transfection of plasmids encoding indicated S protein, ACE2, TMPRSS2, and GFP. 'GFP-only' did not include S plasmid. Images were captured 24 hr after transfection. Major syncytia are traced with a yellow line. **C.** (Left) Analytical SEC chromatograms of the S Wuhan-Hu-1, ΔN135, and ΔN25 S variants in cell culture supernatants on an SRT-10C SEC-500 15 cm column. The T and M indicate the trimer and monomer peaks, respectively. (Right) SEC chromatograms of the purified trimers of the S variants. **D.** Main melting event temperatures (Tm$_{50}$) of the S protein variants. Data are represented as mean + SD of n = 3 replicates.

protein did not yield syncytia (Fig 1B). On the contrary, major syncytia formation was observed with the Wuhan-Hu-1 S protein. Likewise, when cells were transfected with either one of the two variant S proteins, clear syncytia were visible (Fig 1B). Next, we determined whether the ΔN25 and ΔN135 spikes allowed entry into HEK293T cells stably expressing human ACE2 and TMPRSS2. To this end, we generated Spike-pseudotyped, HIV-based lenti-viral particles carrying a luciferase reporter gene. In line with the cell-cell fusion assays, we observed that both variant spikes gave signals comparable to Wuhan-Hu-1 S (S2A Fig). These data demonstrate that the variant S proteins remain fully functional despite considerable changes in the NTD.

## Characterization of the ΔN25 and ΔN135 spikes

We designed soluble versions of the variant S proteins and produced them in transiently trans-fected expi293F cells to enable biochemical and structural characterization. To obtain high quality S proteins with reasonable yields, the furin cleavage site was mutated and stabilizing substitutions to proline were added at positions 892, 987, and 942 in the S2 domain [31,32]. The variant spikes were produced at levels comparable to the Wuhan spike in the crude cell culture supernatant (Fig 1C, left panel). The quaternary structure of the ΔN25 spike was less stable and showed a higher fraction of monomeric S compared to the ΔN135 and Wuhan vari-ants. After purification, only trimeric S proteins remained (Fig 1C right panel). These purified proteins were used for all subsequent experiments. All three S proteins showed the typical

minor melting event at approximately 49°C and a higher main melting event that differed among the spikes. The $Tm_{50}$ of the ΔN25 spike was 2.5°C higher, and that of the ΔN135 spike was 2.5°C lower, as compared to the Wuhan spike (Figs 1D and S3). These results indicate the purified ΔN135 or ΔN25 spike can form a stable trimer that is comparable to the Wuhan-Hu-1 spike.

## Antigenicity of the variant spikes

To investigate the impact of the variant point mutations and deletions on the antigenicity, we measured the binding of a selection of convalescent sera and mAbs to the purified ΔN25 and ΔN135 spikes and compared the binding to the Wuhan-Hu-1 spike. Binding of ten different convalescent sera from patients that were all diagnosed with COVID-19 (BioIVT, S3 Table) were tested using ELISA and revealed that both the ΔN25 and the ΔN135 spikes have reduced binding (S4 Fig). The antigenic assessment with mAbs was performed using biolayer interferometry to measure S protein binding to ACE2-Fc and a panel of six SARS-CoV-2 neutralizing antibodies directed against the RBD (S2M11, S2E12, C144, 2–43, S309 and COVA2-15 [2,4,33–35]), three neutralizing antibodies against the supersite of the NTD (2–51, COVA1-22 and 4A8 [4,16,19]) and a non-neutralizing antibody against the lower part of the NTD (DH1055) [36] (Figs 2A and S5A). The NTD-specific antibodies lost all binding to both variant spikes, except for some residual binding of DH1055 to the ΔN135 S protein. These observations indicate the deletions and mutations in ΔN25 and ΔN135 spikes may significantly reorganize the local structures at the binding epitopes of these neutralization mAbs. Interestingly, although ACE2-Fc can bind to ΔN25 and ΔN135 spikes comparable to the Wuhan-Hu-1 spike, both variants also showed reduced sensitivities to the RBD-specific neutralizing antibodies, except S2E12 and S309 antibodies that target the conserved RBD sites. Especially, RBD-specific mAbs responses to the ΔN135 variant RBD were most significantly impacted. These results indicate ΔN135 and ΔN25 still preserved the binding to the host receptor ACE2 but significantly reduced the responses to most of NTD or RBD directed neutralizing antibodies, revealing their prominent increased immune escape potential. The loss of binding to the RBD is most likely caused by the E484K mutation, which is part of the epitopes of the mAbs S2M11, 2–43, C144 and COVA2-15. [37]

Next, we tested the neutralization capacity of ACE2-Fc, COVA1-22, 2–43 and COVA2-15 using HIV-based lentiviral particles equipped with Wuhan-Hu-1, ΔN25 or ΔN135 Spike on HEK293T cells stably expressing human ACE2 and TMPRSS2 (S2B Fig). In line with the binding observed in the biolayer interferometry experiments, we show that the ΔN25 and ΔN135 spikes remain fully sensitive to inhibition by a soluble form of the receptor (ACE2-Fc). COVA1-22 and 2–43, which show modest inhibition of particles equipped with Wuhan-Hu-1 S in this experimental setup, were even less effective against the two variant spikes. The most dramatic effect was observed for COVA2-15, which showed full neutralization of Wuhan-Hu-1 S, but negligible neutralization of the two variant spikes (S2B Fig).

## Shift in signal peptide cleavage site and subsequent loss of disulfide

In SARS-CoV-2 S, a conserved cysteine Cys15 is present near the N-terminus of the mature protein and forms a disulfide bond with Cys136. Only in the case of the branch of coronaviruses that includes SARS-CoV-2 S, the cysteine is located almost directly adjacent, two residues away from the signal peptide (SP) cleavage site (Fig 3). ΔN135 contains a P9L mutation in the signal peptide, three residues away from the predicted cleavage site (Fig 2B). The P9L mutation would cause a cleavage site shift downstream to Cys15 according to SignalP-6.0 prediction [38]. Although a cleavage site shift has been suggested due to the P9L mutation [28], it has not

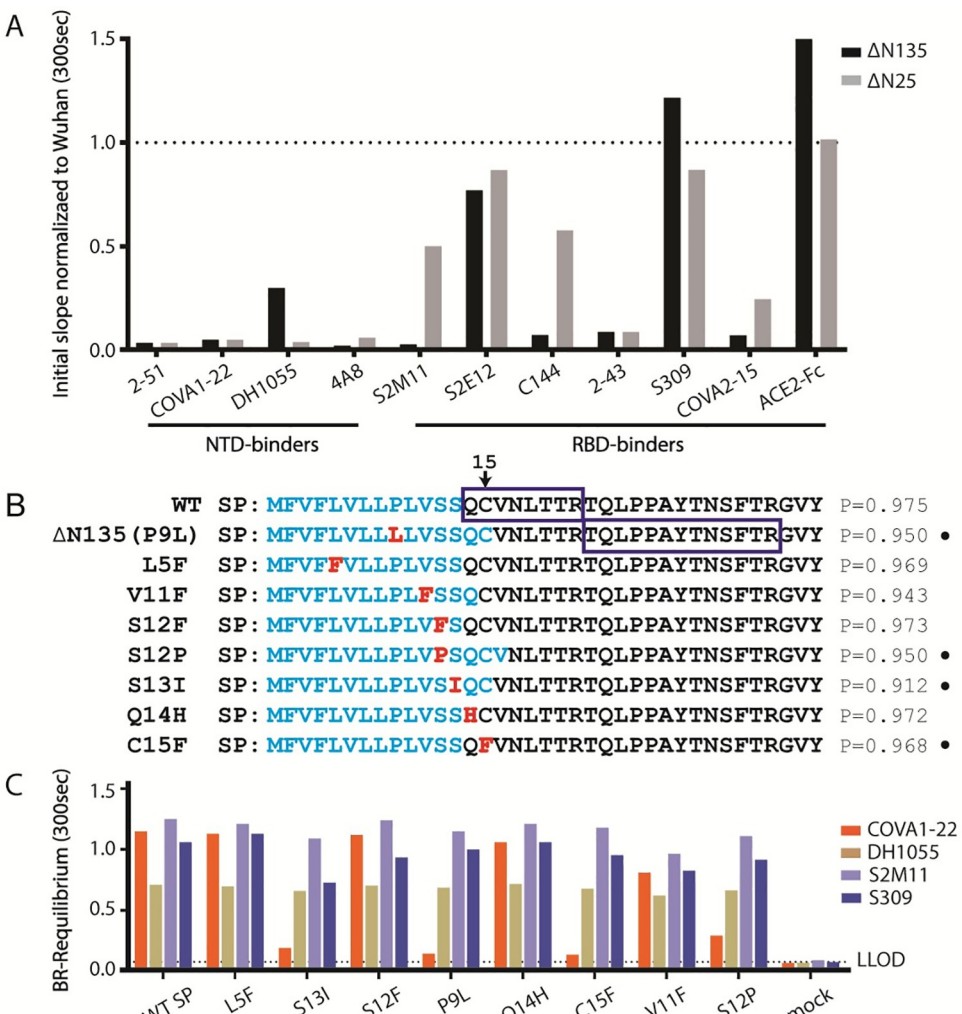

**Fig 2. Impact of NTD deletions and mutations on antigenicity. A.** Binding of NTD- and RBD-specific mAbs and ACE2-Fc to purified ΔN135 and ΔN25 S trimer measured with BioLayer Interferometry, showing the initial slope $V_0$ at the start of binding, normalized to that of the Wuhan variant (dashed line). **B.** The predicted signal peptide (SP) of the WT or mutant S is shown in blue bold characters with the probability (P) of the SP cleavage site as predicted by SignalP 6.0. The mutations are highlighted in red. Most N-terminal peptide detected using mass spectrometry is boxed. The peptides QCVNLTTR, VNLTTR or NLTTR are not found for the ΔN135 S. Mutations that result in loss of Cys15-Cys136 are highlighted with filled circles. **C.** Binding of mAbs to the S trimers with WT SP or different mutations in or just after the SP. Binding was measured with Biolayer Interferometry using Octet on crude cell culture supernatants. The R equilibrium calculated at 300 seconds is shown as bars. LLOD: lower limit of detection.

been proven and therefore we performed liquid chromatography-mass spectrometry (LC-MS/MS) from a tryptic digest of purified Wuhan-Hu-1 and the ΔN135 S protein to determine the N-terminal residue of the mature proteins. We found that, in line with published observations [30], the Wuhan-Hu-1 S protein was cleaved after position 13 (Figs 2B and S6). In contrast, for the ΔN135 S protein, no peptides were detected up to N-terminal residue 22 which would correspond to a truncation of 8 N-terminal residues including Cys15 (Fig 2B). Interestingly, in the ΔN135 spike the loss of Cys15 is accompanied by the loss of Cys136 due to the large deletion of residues 136–144. The loss of both cysteines indicates this could be a compensatory mutation since an unpaired cysteine can impact correct folding of the spike. The loss of disulfide $DS_{15-136}$ may consequently impact the structural architecture of the ΔN135 NTD (see below).

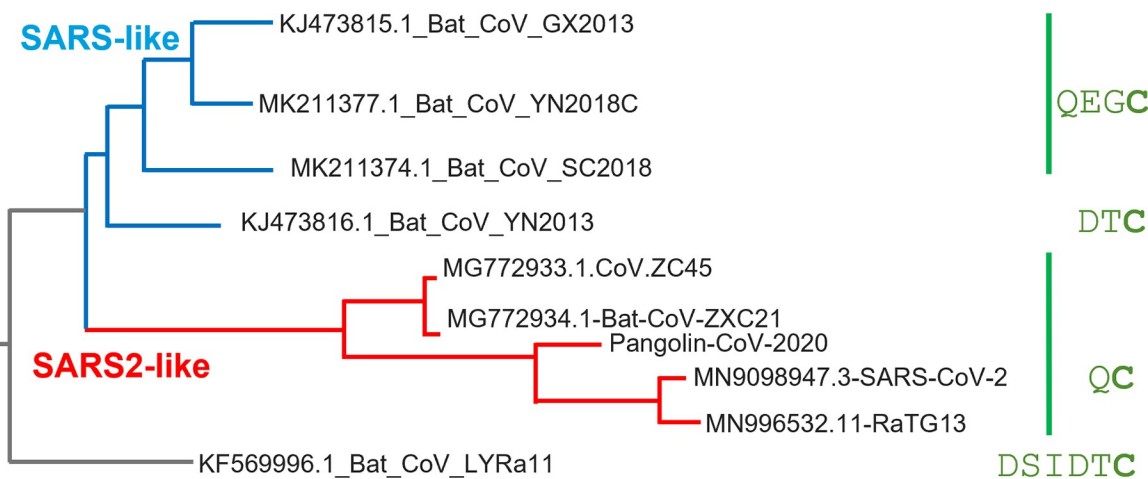

**Fig 3. Evolutionary relationships among SARS1-like coronaviruses (blue) and SARS-CoV-2-like coronaviruses (red) with the N-terminal residues of spike of the indicated branch in green.**

## Spread of the DS$_{15-136}$ breaking mutations

P9L and the previously described S13I [27] cause a shift in signal peptide cleavage, resulting in the loss of Cys15. This SP-shift can be indirectly detected by the loss of binding to mAb COVA1-22 (Figs 2C and S5B) which binds to the NTD N-terminus. A panel of common SP mutations, including P9L and S13I, was evaluated for mAb COVA1-22 binding to investigate the occurrence of both the signal peptide cleavage shift and concomitant loss of DS$_{15-136}$. Apart from P9L, S13I and C15F, only S12P resulted in reduced COVA1-22 binding which agreed with the predicted signal peptide cleavage shift according to the SignalP-6.0 software [38] (Figs 2C and S5B). Meanwhile, these SP-shifts did not interfere with the binding of other mAbs like DH1055, S2M11, or S309.

NTD is a hotspot for deletions in the S protein, and the same deletions keep evolving on independent branches of the phylogenetic tree of S (Figs 3 and 4A). The same is true for the loss of the Cys15-Cys136 disulfide (ΔDS$_{15-136}$) which can occur via mutation or deletion of either of the two cysteine residues (Fig 4B). S13I and P9L are the most frequent causes for the loss of Cys15 via the cleavage site shift mechanism, but direct mutation of Cys15 is also observed (S4 Table). Cys136 is removed only via direct mutation and occurs less often. Approximately half of the lineages with ΔDS$_{15-136}$ have both cysteines removed as in the Russian AT.1 lineage [39] or the C.1.2 lineage [28]. The distribution of the ΔDS$_{15-136}$ variants on the phylogenetic tree of SARS-CoV-2 S (Fig 4C) and the different paths leading to the disulfide loss (S3 Table) suggest that ΔDS$_{15-136}$ could have evolved in multiple lineages independently, and in several cases became dominant within the lineages. Fig 4D shows the most significant incidences of ΔDS$_{15-136}$ in SARS-CoV-2 lineages. Before the Delta variant became dominant and outcompeted many of these lineages, in many cases, percentage of ΔDS$_{15-136}$ showed an ascending trend. After replacement of most of the strains by Delta and subsequently Omicron lineages, once again, ΔDS$_{15-136}$ was reemerging in diverse geographical locations (S5 Table).

## Cryo-EM analysis of variant spikes

To understand the structural impact of the large NTD deletions and the loss of DS$_{15-136}$ (ΔDS$_{15-136}$) in the ΔN135 variant, we solved the Cryo-EM structures of the stabilized ectodomains of both spike variants (Figs 5, S7, and S8 and S3 Table). The overall structures of the

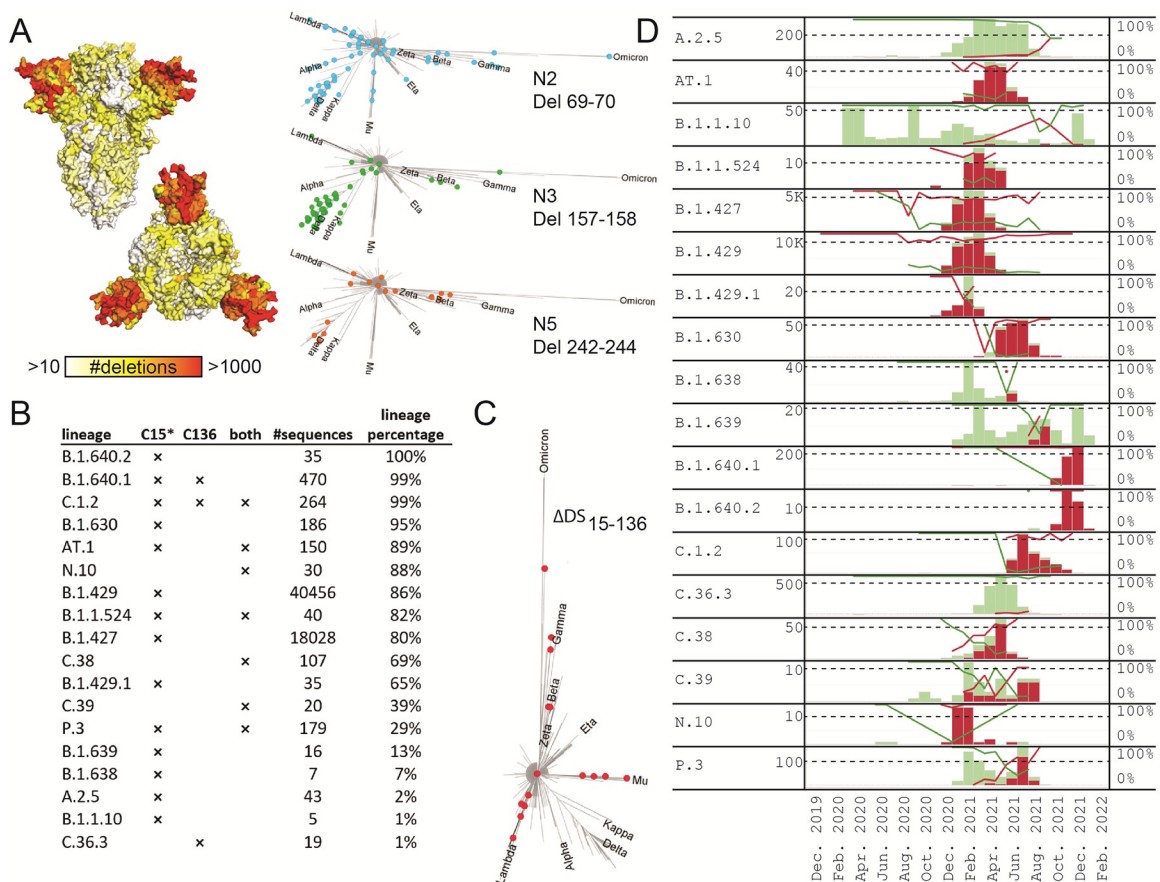

**Fig 4. NTD deletions and occurrences of DS15-136 loss in lineages. A.** Deletion frequency based on GISAID 25 Jan 2022, plotted on spike surface (side and top view). White corresponds to positions with less than 10 deletions registered on GISAID, while red is assigned to positions with more than 1000 deletions observed. In the right panels, deletions in N2, N3 and N5 loops are plotted on the S-protein phylogenetic trees (see Materials and Methods section), with the major variants of concern indicated as reference. Only deletions that were identified in more than 1% of their respective lineage sequences are plotted. **B.** List of lineages containing at least 1% of $\Delta DS_{15-136}$ variants. For each lineage, the mechanism of the $DS_{15-136}$ loss is indicated with a cross, where C15* stands for both a direct C15 mutation or deletion, or signal peptide mediated cleavage site shift. C136 stands for C136 mutation, and "both" indicates both cysteines were removed via any of the possible mechanisms. The fraction of $\Delta DS_{15-136}$ sequences within each lineage was calculated in the last column. **C.** Lineages containing at least 1% of $\Delta DS_{15-136}$ variants plotted on the phylogenetic tree of SARS-CoV-2 S. **D.** Time evolution of the $\Delta DS_{15-136}$ containing lineages, with variants containing the cystine bridge colored in green and variants without the cystine bridge colored in red. The bars depict lineage counts for each month (left axis), and the lines–percentages within each lineage of both sub-variants (right axis).

$\Delta N135$ and $\Delta N25$ trimers are the same as that of the Wuhan spike with the D614G mutation except for the loops in the NTD [40]. $\Delta N135$ variant acquires predominantly the 1-RBD up conformation (73% 1-up, 23% down) (S3 Table and S7 Fig). Deletion of N1 results in loss of $DS_{15-136}$ and exposes a hydrophobic patch which contributes to a large reorganization of the NTD loops. The conserved N2 loop has completely shifted position and occupies the space of the deleted N1 loop (Fig 5B). The deletion of one of the strands of the N3 beta-hairpin destroys the 3-strand β-sheet $\beta_{N3N5}$. As a result, N3 completely shifts and occupies the space of the deleted N1 loop. Finally, the deletion in N5 and the loss of the secondary structure of $\beta_{N3N5}$ results in a shift of N5 to the space previously occupied by N2 and N4 shifts away from the other loops. The loss of $DS_{15-136}$ and $\beta_{N3N5}$ due to the deletions in N1, N3 and N5 causes a dramatic remodeling of the N2, N3, N4 and N5 loops that include the NTD supersites (Fig 5B) and a reduced stability of the spike (Fig 1D). From the $\Delta N25$ spike dataset, one stable class

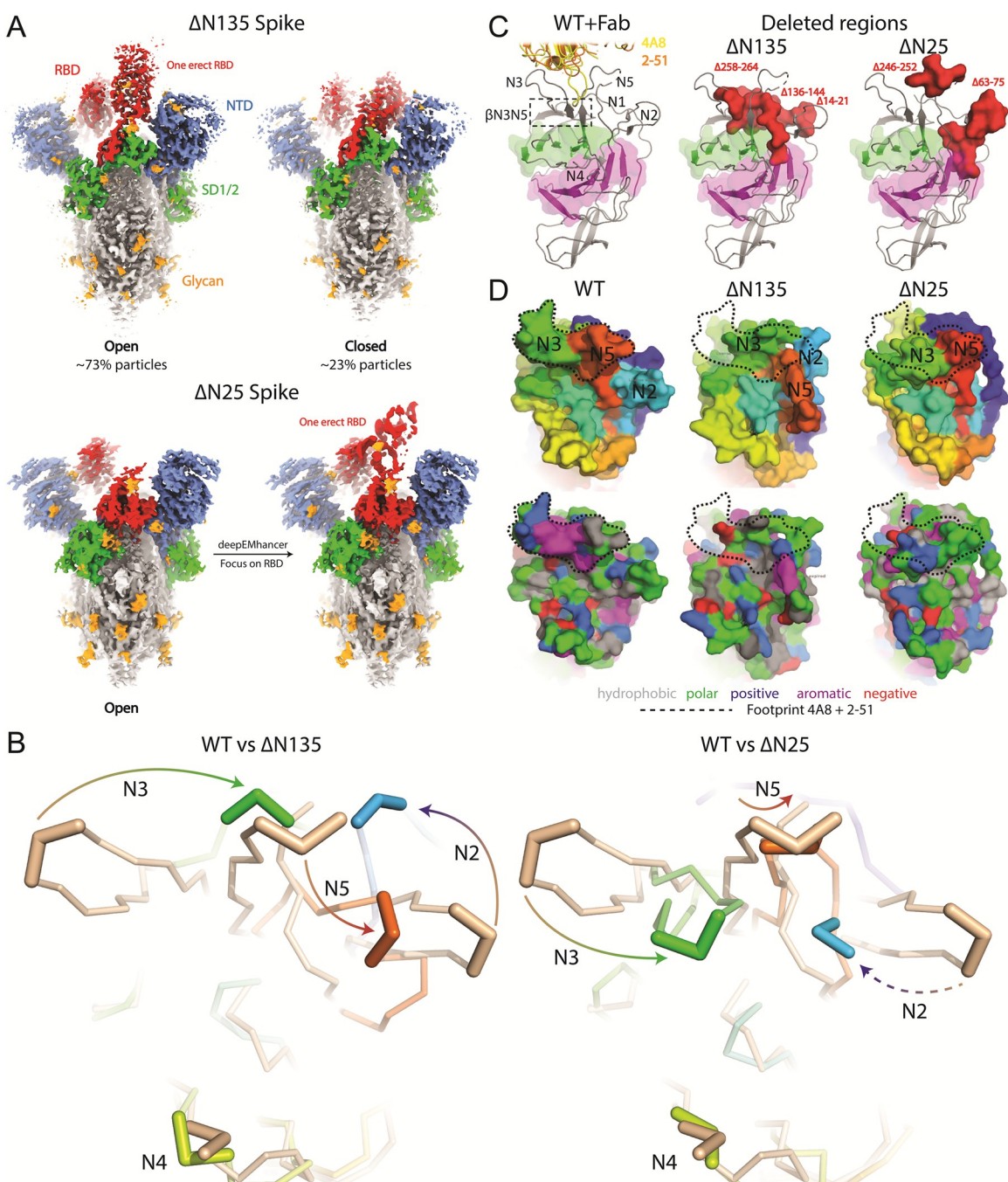

**Fig 5. Conformational plasticity of NTD in spike variants. A.** Cryo-EM structures of ΔN135 and ΔN25 Spikes. **B.** Superposition of ribbon representation of the reference (in wheat) and variant NTDs. Arrows indicate rearrangement of the loops as a result of the deletions. The dashed arrow indicates the complete deletion of the N2 loop in the ΔN25 variant. **C.** Left panel: sideview of the NTD with the two sheets of the galectin-fold in green and magenta and indicated N-loops. The βN3N5 sheet on top of the galectin-fold is boxed with a dashed line. As a reference for the NTD supersite, structures of Fabs 4A8 (PDBID 7C2L) and 2–51 (PDBID 7L2C) are indicated in yellow and orange ribbons. In the middle and right panels, the deleted amino acids are depicted for the ΔN135 and ΔN25 spikes in red as space filling representation. **D.** Surface representation of the NTD is colored by N-loops (upper panel) and the properties of the residue types (lower panel). Dashed contours indicate the joint footprint of mAbs 4A8 plus 2–51 on the reference spike (PDBID 7C2L). The epitope contour was also plotted over the variant NTDs as guidance to indicate the changes introduced by the deletions.

with one RBD-up was able to be refined into high resolution (Figs 5A and S8). The ΔN25 spike has a 7-residue deletion in the N5 loop typical for the C.37 lineage [41]. As a result of this deletion and the complete loss of the N2-loop due to the large 13-residue deletion of residues 63–75, the N5-loop shifts towards the N2 and N1 loops and concomitantly, the N3-loop shifts to a position previously occupied by N5 (Fig 5B). As a result of the deletions and N-loop shifts, the 3-strand β-sheet formed by N3 hairpin and N5 ($\beta_{N3N5}$) on top of the galectin-fold is lost and as a result, the N4-loop is shifted away from the other loops. The structural overlay with ΔN135 and ΔN25 at NTD reveals the deletions do not only change the architecture, but also alter the surface properties, which explains the loss of interaction with antibodies 4A8 and 2–51 for which the complex structures were solved [16,19] (Figs 2A, 5C and 5D). The deletions and remodeling of N2, N3, N4 and N5 result in major antigenic changes in the NTD supersite (Figs 2, 5B–5D, and S9).

## Discussion

The rapid global spread of SARS-CoV-2 leads to emergence of variants with either higher transmissibility or decreased recognition by protective immune response. The NTD undergoes rapid antigenic drift and accumulates a larger number of mutations and especially deletions [42,43]. In this study, we describe two spike variants, one from Peru and one from Brazil with typical point mutations in the RBD but extensive and rare deletions in the NTD (Fig 1). Since the observed deletions are extensive, we examined folding and function of the variant spikes and investigated their structural impact. Both spikes showed robust expression and maintained fusogenicity, and the purified soluble proteins showed comparable thermostability and ACE2 binding (Figs 1B, 1C and S3). As a result of deletions, both spikes show complete loss of antibody binding to the NTD supersite (Figs 2 and 4B). Additionally, the mutations in the ΔN135 spike impacted binding of the majority of the RBD specific antibodies (Fig 2A). As expected also binding of different convalescent sera was reduced to the ΔN25 and ΔN135 spikes compared to the Wuhan-hu-1 spike (S4 Fig and S2 Table). The ΔN25 variant derived from the C.37 lineage with a large 7-residue deletion in the N5 loop [41] acquired an additional 13-residue deletion in the N2 loop. The ΔN135 variant belonging to the B.1.1.294 lineage acquired three large deletions: a 9-residue deletion in N3, a 7-residue deletion in N5, and an N-terminal deletion due to the signal peptide cleavage shift, also leading to the $DS_{15-136}$ loss. Structural analysis of the proteins using Cryo-EM showed that the overall fold of the spikes was maintained and the galectin-fold of the NTD remained intact despite the large deletions and loss of the disulfide bridge (Figs 5 and S7). However, the loops that constitute the NTD supersite were completely remodeled or relocated in both proteins (Fig 5D), which explains the dramatic changes to the NTD antigenicity profile (Fig 2). In the ΔN25 spike complete deletion of the N2 and partial deletion of N5 loop results in large shift of the N3 and N4 loops. In the ΔN135 spike, N2 and N3 move to the position of the deleted N1 and N4 moves away from the other loops. The relocation of the loops was enabled by the loss of the $\beta_{N3N5}$ β-sheet due to deletion of the N3 β-hairpin and the deletion in the N5 loop.

Aside of the extensive loop deletions, the virus can remodel the NTD supersite by shifting its signal peptide cleavage site with the P9L point mutation. We experimentally verified that the mutation causes a longer truncation of the N-terminus by Mass spectrometry of tryptic digests, loss of binding to mAb COVA1-22 specific for the NTD N-terminus and by Cryo-EM structure determination (Figs 2, 4 and 5). S13I and to a lesser extend S12P also cause the peptide cleavage shift (Fig 2B and 2C) which has been demonstrated for S13I in the B.1.427/B.1.429 variant [27]. Next to the direct mutation or deletion of one of the cysteines, the signal peptide mutations constitute an additional mechanism via which $\Delta DS_{15-136}$ can occur [27].

The mutations that shift the cleavage site, together with the Cys15 and Cys136 mutations and deletions were used to identify $\Delta DS_{15\text{-}136}$ variants in the GISAID database (S4 Table and Fig 4B and 4C). Although these modifications are relatively rare, $\Delta DS_{15\text{-}136}$ is widespread both geographically and in terms of occurrences on the phylogenetic tree of S. This escape mechanism arose independently in different geographical locations and even became dominant in some lineages until Delta replaced most other variants around the world. However, in the midst of the ongoing Omicron wave, Colson et al. [44] reported an emergence of a new concerning variant (B.1.640.2) in Southern France, probably of Cameroonian origin which also evolved the $\Delta DS_{15\text{-}136}$ feature. $\Delta DS_{15\text{-}136}$ was also observed in a globally spreading Omicron sublineage BA.2.3.21, initially spreading in Philipinnes and United States, but now also observed in Australia and multiple South-East Asian countries.

In the last three years, the NTD domain of the SARS-CoV-2 spike has been confirmed as a hotspot for deletions [42,43]). Within NTD, deletions are further clustered around a few sites: residues 69–70, 141–143, 156–159, and 242–245. Deletions at these sites recur independently in large number of unrelated lineages, as depicted in the phylogenetic trees of SARS-CoV-2 S (Fig 4A). The large capacity for deletions in N2, N3 and N5 loops together with the ability to remove N1 with the $\Delta DS_{15\text{-}136}$ mechanism to further rearrange all surrounding loops allows the virus to completely remodel the NTD supersite (Figs 4–6, S7, and S8). Moreover, the mechanism of reshaping the loops via $\Delta DS_{15\text{-}136}$ seems to have evolved independently in multiple branches of the SARS-CoV-2 phylogenetic tree, suggesting this escape mechanism may also play a role in the future variants of concern. The NTD loop length variation has been suggested to play a role in optimization of protein function, antigenic variation and adaptation to a new host [42].

As collective immunity to the virus increases, immune evasion will likely become an important fitness advantage, as recently observed for the Omicron variant. It is likely that escapes via structurally tolerated large deletions and/or the $\Delta DS_{15\text{-}136}$ mechanism will occur again when selection based on immune evasion continues. In fact, deletions of the loops are already firmly incorporated in the Delta and Omicron lineages. $\Delta DS_{15\text{-}136}$ has also been registered in these variants of concern. When analyzed locally (S5 Table), at the end of the Delta wave Delta lineages in Sweden and Chile started to develop $\Delta DS_{15\text{-}136}$. With the rise of Omicron these lineages were eventually outcompeted, but as the new wave progressed multiple cases of Omicron BA.1, BA.2 and BA.5, $\Delta DS_{15\text{-}136}$ have also been registered in local outbreaks around the world. One Omicron lineage in particular (BA.2.3.1) containing the $\Delta DS_{15\text{-}136}$ feature has already spread globally. With increasing global immunity, the escape mechanisms that are currently rare, should be closely monitored and it would be important to understand the constraints of the NTD erosion and the balance between NTD function and structural integrity.

## Material and methods

### Ethics statement

The clinical study ENSEMBLE (COV3001, NCT04505722) is a multinational study conducted in 225 different sites, located in eight different countries (Argentina, Brazil, Chile, Colombia, Mexico, Peru, South-Africa, and the USA), and therefore many different Ethics Committees (>80) were involved. We hereby state that approval was granted by all Ethics Committees/Institutional Review Boards. A list of ethics committees and boards with corresponding approval numbers is provided as S1 Text.

### Ethical conduct of the study

This study was conducted in accordance with the ethical principles that have their origin in the Declaration of Helsinki and that are consistent with Good Clinical Practices and applicable

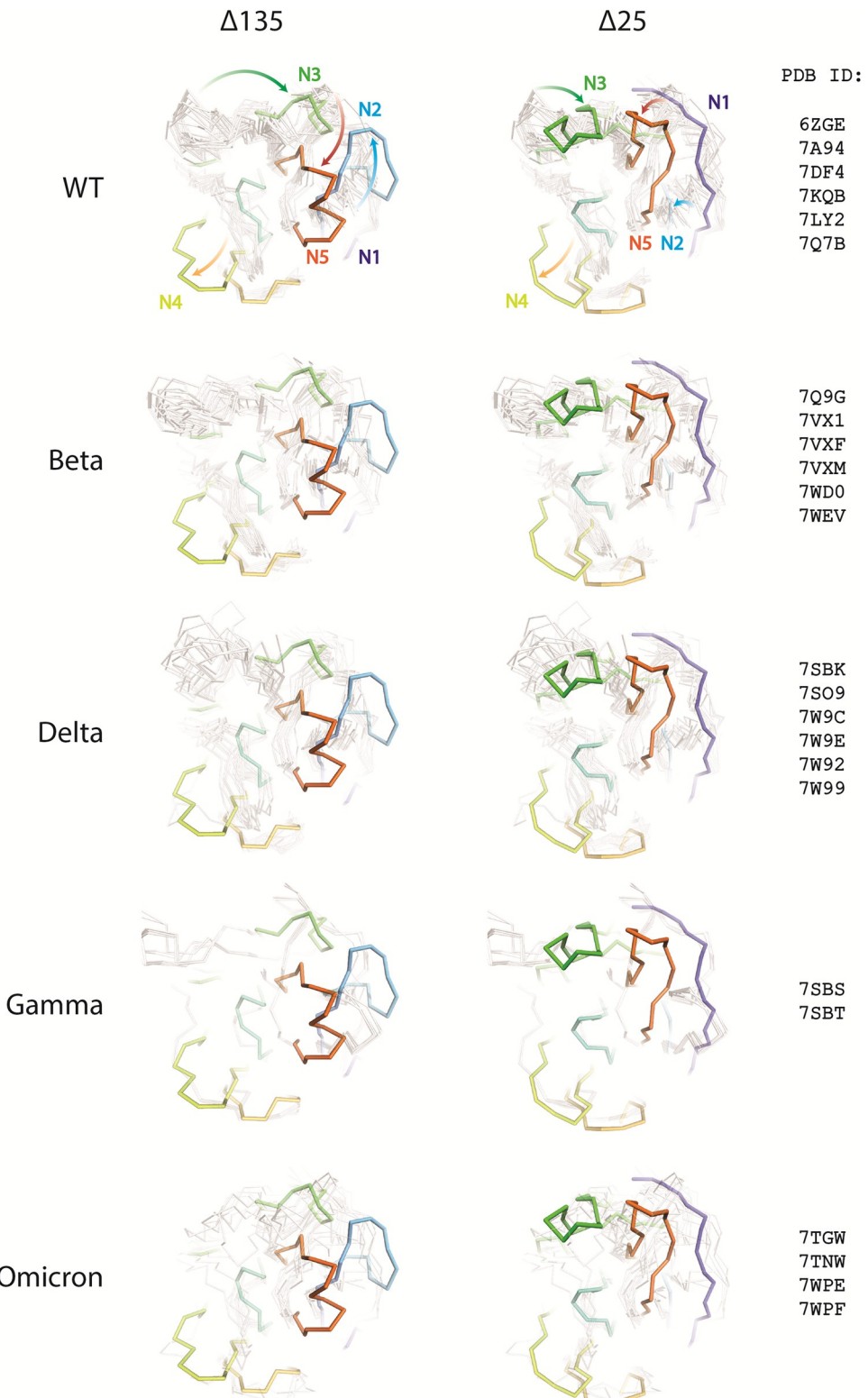

**Fig 6. Superposition of ribbon representation of ΔN135 (left) and ΔN25 (right) NTD compared with Wuhan (WT) and other variants with colors and view as in Fig 4B.** Structures from individual variant were selected based on the completeness of supersite loops. Arrows indicate rearrangement of the loops as a result of the deletions.

regulatory requirements. Known instances of nonconformance were documented and are not considered to have impacted the overall conclusions of this study.

## Participant information and consent

Participants or their legally acceptable representatives provided their written consent to participate in the study after having been informed about the nature and purpose of the study, participation/termination conditions, and risks and benefits of treatment. Participants were to be given sufficient time to read the Informed Consent Form (ICF) and the opportunity to ask questions. Before entry into the study, consent was to be appropriately recorded by means of the participant's personally dated signature. Known instances of nonconformance were documented and are not considered to have impacted the overall conclusions of this study.

## Clinical samples

Nasal swab specimens from SARS-CoV-2 RT-PCR confirmed cases, selected to be as close as possible to the onset of symptoms and having a SARS-CoV-2 viral load >200 copies/ml, were selected for sequencing. Molecular confirmation of SARS-CoV-2 infection and viral load quantification was performed using the Abbott RealT*ime* SARS-CoV-2 RT-PCR at the Virology Laboratory of the University of Washington, Department of Laboratory Medicine and Pathology (UW Virology, Seattle, US).

## Next-generation sequencing

SARS-CoV-2 whole genome sequencing was performed by UW Virology using the Swift Biosciences SNAP Version 2.0 assay (Integrated DNA Technologies) run on the Sciclone G3 NGSx iQ workstation [45,46]. Briefly, RNA was extracted from 200μL of swab VTM using the MagNA Pure 96 System (Roche). 11μL RNA is incubated with 1μL 50mM random hexamer (Thermo Fisher) and 1μL 10mM dNTP (Thermo Fisher) at 65°C for 5 min and cooled to 10°C. cDNA was generated using the SuperScript IV First-Strand Synthesis System by adding 4μL 5X SSIV Buffer, 1μL RNase inhibitor, 1μL 0.1M DTT, and 1μL SSIV RT (Thermo Fisher) and incubating at 23°C for 10 min, 50°C for 30 min, and 80°C for 10 min before cooling to 10°C. 10μL of cDNA was used as input for the SNAP assay (then Catalog # CovG1 V2-96, now xGen SARS-CoV-2 Amplicon Panel), following manufacturer's recommended protocol and sequenced on the NextSeq2000 using a P2 300 cycle kit (150x150bp run). Genomes from Swift SNAP libraries were assembled using a custom pipeline (https://github.com/greninger-lab/covid_swift_pipeline) described previously [45]. Reads adapter and quality trimmed using BBDuk (https://jgi.doe.gov/data-and-tools/bbtools), aligned to the Wuhan–Hu-1 reference genome (NC_045512.2), and trimmed of PCR primers using Primerclip (https://github.com/swiftbiosciences/primerclip). Consensus genomes and variants were called using BCFtools. Genome acceptability criteria included >1 million raw reads, >750× mean genome coverage, >1000× mean spike gene coverage, 100% of spike gene with at least 200× coverage, and <10% missing data (Ns) in the final consensus sequence. A minimum of 6× depth per base was required to call consensus sequence.

## Protein expression and purification

Plasmids corresponding to the SARS-CoV2 S variant proteins truncated after residue 1208 and with stabilizing substitutions A892P, A942P, D614N and V987P and a furin cleavage site knock out (R682S, R685G) were synthesized and codon-optimized at GenScript (Piscataway, NJ 08854). The constructs were cloned into pCDNA2004 or generated by standard methods

widely known within the field involving site-directed mutagenesis and PCR and sequenced. The expression platform used was the Expi293F cells. The cells were transiently transfected using ExpiFectamine (Life Technologies) according to the manufacturer's instructions and cultured for 6 days at 37˚C and 10% CO2. The culture supernatant was harvested and spun for 5 minutes at 300 g to remove cells and cellular debris. The spun supernatant was subsequently sterile filtered using a 0.22 μm vacuum filter and stored at 4˚C until use. S trimers were purified using a two-step purification protocol by Lentil Lectin from *Galanthus Nivalis* (Vector labs, catalog AL-1243., followed by by size-exclusion chromatography using a HiLoad Superdex 200 16/600column (GE Healthcare).

## Antibodies and reagents

ACE2-Fc was made as described [47]. Kidney international. For 2–51, DH1055, 4A8, S1M11, S2E12, C144, 2–43 and S309 the heavy and light chain were cloned into a single IgG1 expression vector to express a fully human IgG1 antibody. Antibodies were produced by transfecting the IgG1 expression constructs using the ExpiFectamine 293 Transfection Kit (ThermoFisher) in Expi293F (ThermoFisher) cells according to the manufacturer specifications. Purification from serum-free culture supernatants was done using mAb Select SuRe resin (GE Healthcare) followed by rapid desalting using a HiPrep 26/10 Desalting column (GE Healthcare). The final formulation buffer was 20 mM NaAc, 75 mM NaCl, 5% Sucrose pH 5.5. COVA1-22 and COVA2-15 have been kindly provided by Marit van Gils.

## SEC and analytical SEC

Plasmids coding for recombinant SARS-CoV-2 S protein ectodomains were expressed in Expi293F cells using ExpiFectamine (Life Technologies) according to the manufacturer's instructions and cultured for three days at 37˚C and 10% CO2. To assess S protein expression with analytical SEC, sterile-filtered crude cell culture supernatant was analyzed by analytical size exclusion chromatography (SEC) using an ultra-high-performance liquid chromatography system (Vanquish, Thermo Scientific). 20 μl of the cleared crude cell culture supernatants was applied to an SRT-C SEC-500 15 cm column (Sepax Cat. #235500–4615) with the corresponding guard column (Sepax Cat. #231300–4605) equilibrated in running buffer (150 mM sodium phosphate, 50 mM NaCl, pH 7.0) at 0.35 ml/min. The OD280 UV trace of the supernatant samples were analyzed using Chromeleon 7.2.8.0 software package. The signal of supernatants of non-transfected cells was subtracted from the signal of supernatants of S transfected cells. Of the purified proteins 10 μg was injected on an SRT-C C-500A 15 cm column (Sepax Cat#235500–4615) with the corresponding guard column (Sepax Cat. #231300–4605) and data were analyzed using Astra 7.3 software package.

## Differential scanning fluorometry (DSF)

20 μg of purified protein in 90 μl PBS pH 7.4 (Gibco) was mixed with 10 μl of 100 times diluted SYPRO orange fluorescent dye (5000 x stock, Invitrogen Cat# S6650) in a 96-well optical qPCR plate. 30 μl of the mixture was then added to a MicroAmp Fast Optical 96-well plate (Thermo Fisher Scientific Cat# 4346906) and the plate was covered with MicroAmp Optical Adhesive Film (Thermo Fisher Scientific Cat# 4311971). A negative control sample containing only the dye was used for reference subtraction. The measurement was performed in a qPCR instrument (Applied Biosystems ViiA 7) using a temperature ramp from 25–95˚C with a rate of 0.015˚C per second. Data was collected continuously. The first derivative was plotted as a function of temperature. The melting temperature corresponds to the lowest point in the curve.

## BioLayer Interferometry (BLI)

The antibodies were immobilized at a concentration of 10 μg/ml on anti-hIgG (AHC) sensors (Sartorius Part No.18-5064) in 1x kinetics buffer (FortéBio cat#18–1105) in 384 well black tilted bottom microplates (FortéBio cat#18–5076). The experiment was performed on an Octet HTX instrument (Pall-FortéBio) at 30˚C with a shaking speed of 1,000 rpm. Sensor hydration was 660 s, immobilization of antibodies 600 s, followed by washing for 300 s and then binding the S proteins at 10 μg/ml in 1x kinetic buffer for 300 s. The data analysis was performed using the FortéBio Data Analysis 12.0 software (FortéBio).

## Cryo-EM Grid Preparation and Data Collection

3.5 μL of 0.8–1.0 mg/ml purified ΔN25 or ΔN135 Spike complex was applied to the plasma-cleaned (Gatan Solarus) Quantifoil 1.2/1.3 holey gold grid, and subsequently vitrified using a Vitrobot Mark IV (FEI Company). Cryo grids were loaded into a Titan Krios transmission electron microscope (ThermoFisher Scientific) with a post-column Gatan Image Filter (GIF) operating in nanoprobe at 300 keV with a Gatan K3 Summit direct electron detector and an energy filter slit width of 20 eV. Images were recorded with Leginon in counting mode with a pixel size of 0.832 Å and a nominal defocus range of -1.8 to -1.2 μm. Images were recorded with a 1.4 s exposure and 40 ms subframes (35 total frames) corresponding to a total dose of ~ 52 electrons per $Å^2$. All details corresponding to individual datasets are summarized in S3 Table.

## Cryo-EM image processing

Dose-fractioned movies were gain-corrected, and beam-induced motion correction using MotionCor2 [48] with the dose-weighting option. The Spike particles were automatically picked from the dose-weighted, motion corrected average images using Relion 3.0 [49]. CTF parameters were determined by Gctf Particles were then extracted using Relion 3.0 with a box size of 440 pixels. The 3D classification and refinement were performed with Relion 3.0 using the binned datasets. One round of 3D classification was performed to select the homogenous particles. Unbinned homogenous particles were re-extracted and then submitted to 3D auto-refinement without symmetry imposed. For ΔN135 Spike, cryoDRGN was performed using the parameters from the last iteration of the 3D auto-refinement. An additional round of no-alignment 3D classification revealed two distinct conformational states of ΔN135 Spike: ~73% of particles adopting an open conformation with one erected RBD was further refined without symmetry imposed; ~23% of particles in the fully closed conformation were further refined with the C3 symmetry imposed. An additional round of no-alignment 3D classification revealed one open state of ΔN25 Spike and was followed by further refinement without symmetry imposed. Focus refinements were performed with soft masks around the NTD, RBD, and body regions. 3D classifications and 3D refinements were started from a 60 Å low-pass filtered version of an ab initio map generated with Relion 3.0. All resolutions were estimated by applying a soft mask around the protein complex density and based on the gold-standard (two halves of data refined independently) FSC = 0.143 criterion. Prior to visualization, all density maps were sharpened by applying different negative temperature factors using automated procedures, along with the half maps, were used for model building. Local resolution was determined using ResMap [50] (S7 and S8 Figs).

## Model building and refinement

The initial template of the Spike complex was derived from a homology-based model calculated by SWISS-MODEL [51]. The model was docked into the EM density map using Chimera

[52] and followed by manually adjustment using COOT [53]. Note that the EM density around the NTD and RBD regions was poor relative to other parts of the model. The NTD and RBD regions were modeled using the unsharpened maps together with the deepEMhancer maps that were calculated with the half maps from the focus refinements. Each model was independently subjected to global refinement and minimization in real space using the module *phenix. real_space_refine* in PHENIX [54] against separate EM half-maps with default parameters. The model was refined into a working half-map, and improvement of the model was monitored using the free half map. Model geometry was further improved using Rosetta. The geometry parameters of the final models were validated in COOT and using MolProbity [55] and EMRinger [56]. These refinements were performed iteratively until no further improvements were observed. The final refinement statistics were provided in S3 Table. Model overfitting was evaluated through its refinement against one cryo-EM half map. FSC curves were calculated between the resulting model and the working half map as well as between the resulting model and the free half and full maps for cross-validation (S9 Fig). Figures were produced using PyMOL (The PyMOL Molecular Graphics System) and Chimera.

## Analytical SEC

An ultra-high-performance liquid chromatography system (Vanquish, Thermo Scientific) and μDAWN TREOS instrument (Wyatt) coupled to an Optilab μT-rEX Refractive Index Detector (Wyatt), in combination with an in-line Nanostar DLS reader (Wyatt), was used for performing the analytical SEC experiment. The cleared crude cell culture supernatants were applied to a SRT-10C SEC-500 15 cm column, (Sepax Cat# 235500–4615) with the corresponding guard column (Sepax) equilibrated in running buffer (150 mM sodium phosphate, 50 mM NaCl, pH 7.0) at 0.35 ml/min. When analyzing supernatant samples, μMALS detectors were offline and analytical SEC data was analyzed using Chromeleon 7.2.8.0 software package. The signal of supernatants of non-transfected cells was subtracted from the signal of supernatants of S transfected cells. When purified proteins were analyzed using SEC-MALS, μMALS detectors were inline and data was analyzed using Astra 7.3 software package.

## Cell-cell fusion assay

A GFP-based cell-cell fusion assay was performed to determine the capability of the variant S protein to mediate membrane fusion. HEK293 cells were transfected with full-length S, human ACE2, human TMPRSS2 and GFP. All proteins were expressed from pcDNA2004 plasmids using Trans-IT transfection reagent according to the manufacturer's instructions. 18hr after transfection, syncytia formation was visualized on an EVOS microscope. Major syncytia in the overlays are traced with a yellow line.

## Pseudotyped virus assays

To show functionality of the Wuhan, ΔN135 and ΔN25 spikes, and to determine their sensitivity to inhibition by ACE2-Fc and a set of monoclonal antibodies, we produced and tested pseudotyped HIV-based lentiviruses as described previously [57]. In brief, pCDNA3.1 expression plasmids encoding for SARS-CoV-2 Spike proteins that were C-terminally truncated by 19 amino acids were used to produce HIV-based lentiviral pseudotyped particles using the ViraPower Lentiviral Expression system (Thermo Fisher Scientific). Spike functionality was shown in a serial dilution of pseudotyped particles in a 2-fold dilution series in assay buffer (Phenol-red free DMEM + 1% nhi-FBS + 1% Pen/strep) on HEK293T cells stably expressing human ACE2 and human TMPRSS2 ('HEK293T_AT'; VectorBuilder; Cat. CL0015) and incubated at 37˚C for 48 hr. Luciferase activity was measured 48 hr post inoculation using the NeoLite

substrate (Perkin Elmer) according to the manufacturer's protocol, with readout performed using an EnSight Plate Reader (Perkin Elmer). For neutralization assays, pseudotyped particle dilutions corresponding to a signal of 1x105 RLU were pre-incubated for 1 hr at room temperature with ACE2-Fc or monoclonal antibodies at indicated final concentrations before being added to HEK293T_AT cells. Luciferase read-out was performed after 48 hr incubation at 37˚C.

## ELISA

Binding of COVID-19 convalescent sera to different spike proteins was measured by ELISA. 50 μl of spike proteins in PBS at a concentration of 1 ug/ml were coated in Perkin Elmer 96-half well area plates O/N at 4˚C. Next day plates were washed three times with 100 μl PBS + 0.1% Tween20, blocked for 1 hr with 100 μl per well of PBS/1% BSA. Subsequently, the plates were incubated for 1 hr with 3-fold serially diluted serum samples in PBS/0.1% Tween20. After washing three times, plates were incubated for 1 hr with 1:5000 HRP conjugated Mouse Anti Human IgG Fcγ fragment (Jackson Cat#209-035-098) in PBS/0.1% Tween20, washed three times again and developed using BM Chemiluminescence ELISA substrate (Roche Cat#11582950001). Luminescence readout was performed using a Perkin Elmer Ensight plate reader.

## GISAID data acquisition and processing

SARS-CoV-2 genome and sample data were downloaded from the GISAID Initiative (https://www.gisaid.org/) database on 25 Jan 2022, and processed by Biovia Pipeline Pilot workflows (BIOVIA, Dassault Systèmes, v 21.2.0.2574, San Diego: Dassault Systèmes, 2020) to transform and standardize the date and country formats, and to retain only human samples. The data are subsequently saved to files with information on individual lineages and individual mutations in Spike protein. The data was further analyzed in Tableau (www.tableau.com) to obtain mutation and lineage frequencies as function of time or location.

Phylogenetic trees in Fig 6A and 6C were created using amino-acid sequences of the S-proteins from GISAID. For each lineage, only one, the most frequent S-protein sequence was used. Only lineages that had 50 or more identical sequences store on GISAID as of 25 Jan 2022 were used. The trees were created using the CLC software.

## Supporting information

**S1 Table. Overview of collected sequences with large deletions in the spike variants from the COV3001 trial.**
(DOCX)

**S2 Table. Sera from COVID-19 patients.**
(DOCX)

**S3 Table. Data collection, reconstruction, and model refinement statistics.**
(DOCX)

**S4 Table. Mutations leading to DS15-136 loss in lineages.**
(DOCX)

**S5 Table. Local occurrences of DS15-136 loss in the Delta and Omicron lineages.**
(DOCX)

**S1 Fig.** A. Schematic representation of SARS-CoV-2 spikes. SP, signal sequence; DS, disulfide bond; S1/S2, S1/S2 Protease cleavage site; SD1, Subdomain 1; SD2, Subdomain2; S2′, S2′ protease cleavage site; HR1, heptad repeat 1; HR2, heptad repeat 2; TM, transmembrane domain. ACE2 binding site on RBD was highlighted. B. Sideview of a spike with the NTD supersites. C. Sequence alignment of NTD. The sequences have been aligned using ClustalW. Positions of the Supersite N-loops and Cys 15,136 were highlighted above the sequence. Predicted signal peptides and the binding epitopes of neutralization mAbs (4A8, and 2–51) were colored in cyan and dark blue, respectively.
(TIF)

**S2 Fig. Infection and neutralization of Spike-pseudotyped lentiviral particles. A.** Infection of HIV-based lentiviral particles pseudotyped with indicated SARS-CoV-2 Spike on HEK293-T_AT cells in a two-fold dilution range. Plotted are relative light units (RLU) based on luciferase expression. **B.** Neutralization assay with the pseudotyped particles of (A) by ACE2-Fc, COVA1-22, 2–43 and COVA2-15 on HEK293T_AT cells at indicated concentrations. Infection is normalized to the signal obtained for the pseudotyped particle in absence of antibody.
(TIF)

**S3 Fig. Differential scanning fluorimetry.** Analysis of melting temperature (Tm) using differential scanning fluorimetry of purified S protein Wuhan-Hu-1 (A), ΔN135 (B) and ΔN25 (C) variants. The first order derivatives are plotted. The experiment was done in triplicate. The Tm is determined as the lowest derivative value representing the Tm50 value.
(TIF)

**S4 Fig. Binding of COVID-19 convalescent sera to Wuhan-Hu-1 (blue line), ΔN25 (orange line), and ΔN135 (red line) spikes measured with ELISA.**
(TIF)

**S5 Fig. BioLayer Interferometry using Octet. A.** Bio-Layer Interferometry curves of the purified Wuhan-Hu-1, ΔN135 and ΔN25 SARS-CoV-2 S proteins binding to ACE2-Fc and a panel of Wuhan-Hu-1 spike binding antibodies. The curves were used to calculate the initial slopes normalized to Wuhan-Hu-1 spike binding plotted in Fig 3. **B.** Impact of SP mutations on Spike NTD antigenicity. Binding of Mabs COVA1-22, DH1055, S2M11 and S309 to the S trimer with D614G, A892P, A942P and V987P substitutions with the wild type signal peptide (wt SP) and with different mutations in or just after the signal peptide, measured with Biolayer Interferometry (BLI) using Octet, showing the binding curves. The curves are labeled in the right lower panel with the mutations present in the signal peptide or just after the signal peptide.
(TIF)

**S6 Fig. Mass spectrometry (MS) analysis of selected peptides. A-B.** ESI-MS (A) and MS/MS (B) spectrum of the most N-terminal peptide observed of wild type S-protein treated with trypsin protease. MS/MS analysis shows the MS plot with the most prominent peaks labelled (left) and a list of the identified fragmented peptides (right). **C-D.** ESI-MS (C) and MS/MS (D) spectrum of the most N-terminal peptide observed of Brazilian variant S-protein treated with trypsin protease. MS/MS analysis shows the MS plot with the most prominent peaks labelled (left) and a list of the identified fragmented peptides (right).
(TIF)

**S7 Fig. Cryo-EM analysis of ΔN135 Spike complex. A.** Flow chart of the cryo-EM data processing procedure. Details can be found in the Materials and methods. **B.** A representative cryo-EM micrograph. **C.** Angular orientation distribution of the particles used in the final

reconstruction. The particle distribution is indicated by different color shades. **D.** Local resolution of the map estimated using the ResMap program and colored as indicated. **E.** Fourier shell correlation (FSC) curve of the structure with FSC as a function of resolution using Relion output. The resolutions are ~3.08 Å and 3.21 Å at the FSC cutoff of 0.143 for the RBD 1-up and all closed Spikes, respectively. **F.** cryoDRGN analysis of the ΔN135 spike revealing the mobility of the RBD highlighted in red arrow.
(TIF)

**S8 Fig. Cryo-EM analysis of ΔN25 Spike complex. A.** Flow chart of the cryo-EM data processing procedure. Details can be found in the Materials and methods. **B.** A representative cryo-EM micrograph. **C.** Angular orientation distribution of the particles used in the final reconstruction. The particle distribution is indicated by different color shades. **D.** Local resolution of the map estimated using the ResMap program and colored as indicated. **E.** Fourier shell correlation (FSC) curve of the structure with FSC as a function of resolution using Relion output. The resolution is ~3.52 Å at the FSC cutoff of 0.143.
(TIF)

**S9 Fig. Cryo-EM densities of the Spike complex. A.** Model validation. Comparison of the FSC curves between model and half map 1 (work), model and half map 2 (free), and model and full map are plotted in red, green, and blue, respectively. **B.** Representative sharpened Cryo-EM density is displayed as mesh at the contour level 15σ. The atomic model with side chains is shown as sticks. **C.** Representative unsharpened Global refinement, Focus Refinement, and deepEMhancer maps around NTD and RBD were shown as surface at 4.5 σ, respectively.
(TIF)

**S1 Text. List of ethics approval numbers.**
(XLSX)

## Acknowledgments

We thank Lam Le, Pascale Boucher and Mandy Jongeneelen for technical support. We would like to thank Marit van Gils for kindly providing COVA1-22 and COVA2-15. The cryo-EM data were collected at NanoImaging Services (San Diego, CA).

## Author Contributions

**Conceptualization:** Xiaodi Yu, Jarek Juraszek, Lucy Rutten, Mark J. G. Bakkers, Johannes P. M. Langedijk.

**Data curation:** Xiaodi Yu, Jarek Juraszek, Lucy Rutten, Mark J. G. Bakkers, Sven Blokland, Jelle M. Melchers, Niels J. F. van den Broek, Annemiek Y. W. Verwilligen, Pravien Abeywickrema, Johan Vingerhoets, Jean-Marc Neefs, Shah A. Mohamed Bakhash, Pavitra Roychoudhury, Alex Greninger.

**Formal analysis:** Xiaodi Yu, Jarek Juraszek, Lucy Rutten, Mark J. G. Bakkers, Sven Blokland, Jelle M. Melchers, Niels J. F. van den Broek, Annemiek Y. W. Verwilligen, Pravien Abeywickrema, Johan Vingerhoets, Jean-Marc Neefs, Shah A. Mohamed Bakhash, Pavitra Roychoudhury, Alex Greninger.

**Supervision:** Lucy Rutten, Mark J. G. Bakkers, Sujata Sharma, Johannes P. M. Langedijk.

**Visualization:** Xiaodi Yu, Jarek Juraszek, Lucy Rutten, Mark J. G. Bakkers.

**Writing – original draft:** Xiaodi Yu, Jarek Juraszek, Lucy Rutten, Mark J. G. Bakkers, Johan Vingerhoets, Sujata Sharma, Johannes P. M. Langedijk.

**Writing – review & editing:** Xiaodi Yu, Jarek Juraszek, Lucy Rutten, Mark J. G. Bakkers, Johan Vingerhoets, Sujata Sharma, Johannes P. M. Langedijk.

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
