## [Decision Letter · Decision Letter 0]

28 Nov 2022

Dear Mr Langedijk,

Thank you very much for submitting your manuscript "Convergence of immune escape strategies highlights plasticity of SARS-CoV-2 spike" for consideration at PLOS Pathogens. As with all papers reviewed by the journal, your manuscript was reviewed by members of the editorial board and by several independent reviewers. In light of the reviews (below this email), we would like to invite the resubmission of a significantly-revised version that takes into account the reviewers' comments.

We cannot make any decision about publication until we have seen the revised manuscript and your response to the reviewers' comments. Your revised manuscript is also likely to be sent to reviewers for further evaluation.

Sincerely,

Florian Krammer, PhD

Academic Editor

PLOS Pathogens

Ron Fouchier

Section Editor

PLOS Pathogens

Kasturi Haldar

Editor-in-Chief

PLOS Pathogens

orcid.org/0000-0001-5065-158X

Michael Malim

Editor-in-Chief

PLOS Pathogens

orcid.org/0000-0002-7699-2064

Reviewer's Responses to Questions

**Part I - Summary**

Reviewer #1: Yu et al. characterize two SARS-CoV-2 variants with deletions in the NTD in the C.37 or B.1.1.129 background, ΔN25 and ΔN135. The ΔN135 variant has in addition to deletions in the NTD a P9L mutation resulting in altered signal peptide cleavage, deletion of a cysteine in spike and consequently loss of the disulfide bound 15-136. The principle of immune escape by mutations close to the signal peptide cleavage side is interesting, has however, also been described by others previously. A weakness of the study is that no data on the actual virus isolates are shown.

Reviewer #2: The authors describe two SARS-CoV-2 isolates with large deletions in the N-terminal domain (NTD) of the viral spike protein. In the manuscript, the authors characterize the effects of these NTD deletions on protein folding and functionality and further determine their effects on antibody binding to the protein. The study is well carried out, the manuscript is clearly written, the data is well presented and the results are solid.

Reviewer #3: In this work, Yu et al., characterize the spike proteins of two SARS-CoV-2 isolates from that carry large NTD deletions. The authors carry out a structural characterization of the proteins and assess their fusogenic capacity.

**Part II – Major Issues: Key Experiments Required for Acceptance**

Reviewer #1: • The authors claim that the spike proteins of the two variants remain fully functional. To support this they only show data from a cell-cell fusion assay with ACE2/TMPRSS2 overexpression and readout 18 hours after transfection. Data from real virus would be interesting to see if replication kinetics are attenuated. Have viruses been isolated or the spike proteins at least been characterized in the context of pseudovirus particles.

• The quaternary structure of soluble ΔN25 spike was less stable compared to wild type and ΔN135. Is there also an impact on stability of membrane bound spike and potentially replication fitness of the virus.

• For antigenic characterization binding of soluble spike proteins to monoclonal neutralizing antibodies is analyzed. Also here data from virus particles would be interesting. Is anything known regarding neutralization of ΔN25 and ΔN135 by polyclonal sera.

Reviewer #2: How do these mutations affect the binding and neutralizing capacity of different monoclonal antibodies, as well as polyclonal sera following infection and vaccination?

Reviewer #3: Detailed characterization of recurrent deletions in the SARS-CoV-2 spike, especially in the NTD have been described (DOI: 10.1126/science.abf6950; DOI: 10.15252/embr.202154322). Authors should put their research in the context of prior data and explain the novelty of their findings and how this contributes to the field.

The authors detect substantial reduction in binding of RBD and NTD mAbs against the Spike variants. It would be ideal to assess the impact of these mutations/deletions on neutralization by the panel of antibodies used in this work, using in vitro assays including live viruses or pseudo viruses expressing the corresponding spikes.

The methods need to be described in detail. As per the current description, it would not be possible to reproduce some of the assays. This is applicable but not limited to the next generation sequencing, protein expression and purification, the BLI, DSF, SEC and fusion assays, which lack a detailed description.

The specific lineage of the SARS-CoV-2 strain from which the sequences were taken should be indicated.

**Part III – Minor Issues: Editorial and Data Presentation Modifications**

Reviewer #1: • Line 146-148: Authors speculate that loss of both cysteines in ΔN135 is a compensatory mutation to avoid an unpaired cysteine. However, the next paragraph describes that deletion of one cysteine alone occurs also in other lineages indicating that the loss of Cys136 could also be a coincidence.

• Line 408: Please add reference correctly

• Line 450: Please change “Brazilian Spike”

• Figure S4B+C: Please add label to axis

Reviewer #2: To more clearly highlight the impact of these mutations on antibody binding and potential escape sites, it could be useful to classify tested mAbs according to their epitopes (NTD supersite, outside supersite, RBD-binding antibodies into class I, II, III and IV) and include representatives covering each of the groups; and show the effect on these deletions on the binding of each group.

it would be easier to visualize and more informative to plot the prevalence of these deletions in different circulating variants in time.

other comments:

line 23, remove "all" from " target for all"

line 26, remove " domain" form "NTD domain", as it's redundant

line 28, "we observed" instead of " we observe"

line 29, "in the loss" instead of " in loss"

line 38, replace "sole" with "main"

Reviewer #3: Line 56. ‘…obtained from samples OF infected…’

Line 62. ‘in vitro’ should be in italics

Line 329. Please rephrase ‘variants with the cystine bridge absent colored in red’ � ‘variants without the cystine bridge colored in red’

PLOS authors have the option to publish the peer review history of their article (what does this mean?). If published, this will include your full peer review and any attached files.

Reviewer #1: No

Reviewer #2: No

Reviewer #3: No
---

## [Decision Letter · Decision Letter 1]

21 Mar 2023

Dear Mr Langedijk,

We are pleased to inform you that your manuscript 'Convergence of immune escape strategies highlights plasticity of SARS-CoV-2 spike' has been provisionally accepted for publication in PLOS Pathogens.

Best regards,

Florian Krammer, PhD

Academic Editor

PLOS Pathogens

Ron Fouchier

Section Editor

PLOS Pathogens

Kasturi Haldar

Editor-in-Chief

PLOS Pathogens

orcid.org/0000-0001-5065-158X

Michael Malim

Editor-in-Chief

PLOS Pathogens

orcid.org/0000-0002-7699-2064

Reviewer Comments (if any, and for reference):

Reviewer's Responses to Questions

**Part I - Summary**

Reviewer #1: Thank you for addressing my comments.

Reviewer #2: The reviewer would like to thank the authors for incorporating the comments and has no further suggestions.

Reviewer #3: The authors have addressed the reviewer's comments adequately. No additional comments/edits.

**Part II – Major Issues: Key Experiments Required for Acceptance**

Reviewer #1: (No Response)

Reviewer #2: (No Response)

Reviewer #3: NA

**Part III – Minor Issues: Editorial and Data Presentation Modifications**

Reviewer #1: Line 473: After adding the reference correctly, please remove the journal name "Kidney international".

Reviewer #2: (No Response)

Reviewer #3: NA

PLOS authors have the option to publish the peer review history of their article (what does this mean?). If published, this will include your full peer review and any attached files.

Reviewer #1: No

Reviewer #2: No

Reviewer #3: **Yes: **Juan Manuel Carreño

---

## [Editor Report · Acceptance letter]

27 Apr 2023

Dear Mr Langedijk,

We are delighted to inform you that your manuscript, "Convergence of immune escape strategies highlights plasticity of SARS-CoV-2 spike," has been formally accepted for publication in PLOS Pathogens.

Best regards,

Kasturi Haldar

Editor-in-Chief

PLOS Pathogens

orcid.org/0000-0001-5065-158X

Michael Malim

Editor-in-Chief

PLOS Pathogens

orcid.org/0000-0002-7699-2064